# Body Transformer: Leveraging Robot Embodiment for Policy Learning

Carmelo Sferrazza     Dun-Ming Huang     Fangchen Liu     Jongmin Lee     Pieter Abbeel

UC Berkeley

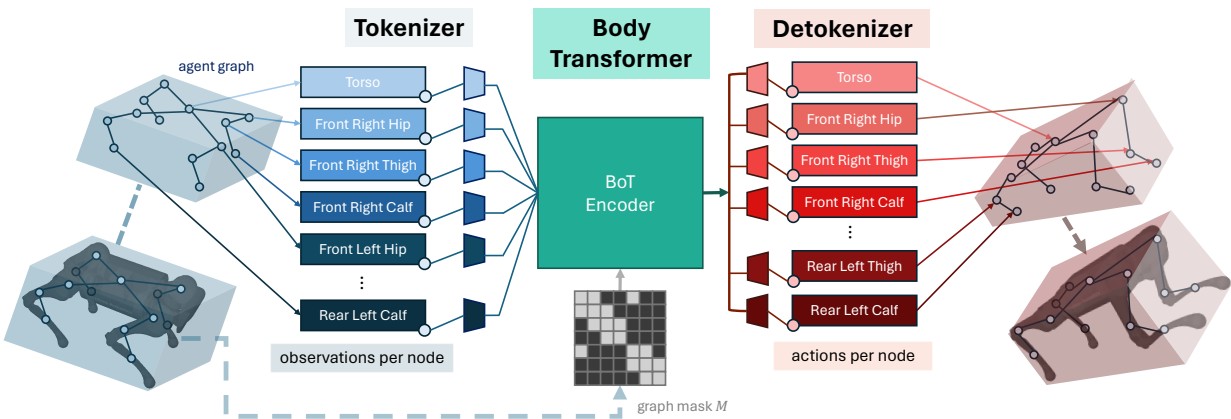

Fig. 1: **Body Transformer** (BoT) is an architecture that considers physical agents as graphs of sensors and actuators as nodes, and edges reflecting the structure of the robot body. BoT leverages masked attention as a simple but flexible mechanism to provide a body-induced bias to the policy. The figure presents the overall schematic of our architecture, exemplified on a Unitree A1 robot.

*Abstract*—**In recent years, the transformer architecture has become the de facto standard for machine learning algorithms applied to natural language processing and computer vision. Despite notable evidence of successful deployment of this architecture in the context of robot learning, we claim that vanilla transformers do not fully exploit the structure of the robot learning problem. Therefore, we propose Body Transformer (BoT), an architecture that leverages the robot embodiment by providing an inductive bias that guides the learning process. We represent the robot body as a graph of sensors and actuators, and rely on masked attention to pool information throughout the architecture. The resulting architecture outperforms the vanilla transformer, as well as the classical multilayer perceptron, in terms of task completion, scaling properties, and computational efficiency when representing either imitation or reinforcement learning policies. Additional material is available at https://bodytransformer.site.**

## I. INTRODUCTION

For most of their correcting and stabilizing actions, physical agents exhibit motor responses that are spatially correlated with the location of the external stimuli they perceive [9]. This is the case of a surfer, where the lower body, i.e. feet and ankles, is mostly responsible for counteracting the imbalance induced by the wave under the board [23]. In fact, humans present feedback loops at the level of the spinal cord's neural circuitry that are specifically responsible for the response of single actuators [24].

Corrective localized actuation is a main factor for efficient locomotion [5]. This is particularly important for robots too, where however, learning architectures do not typically exploit spatial interrelations between sensors and actuators. In fact, robot policies have mostly been exploiting the same archi-

tectures developed for natural language or computer vision, without effectively leveraging the structure of the robot body.

This work focuses on transformer policies, which show promise to effectively deal with long sequence dependencies and seamlessly absorb large amount of data. The transformer architecture [27] has been developed for unstructured natural language processing (NLP) tasks, e.g., language translations, where the input sequences often map to reshuffled output sequences. In contrast, we propose Body Transformer (BoT), an architecture that augments the attention mechanism of transformers by taking into account the spatial placement of sensors and actuators across the robot body.

BoT models the robot body as a graph with sensors and actuators at the graph's nodes. Then, it applies a highly sparse mask at the attention layers, preventing each node from attending beyond its direct neighbors. Concatenating multiple BoT layers with the same structure leads to information being pooled throughout the graph, thus not compromising the representation power of the architecture.

Our contributions are listed below:
- We propose the BoT architecture, which augments the transformer architecture with a novel masking that leverages the morphology of the robot body.
- We incorporate this novel architecture in an imitation learning setting, showing that the inductive bias provided by BoT leads to I) better steady-state performance and generalization and II) stronger scaling properties.
- We show how BoT improves existing online reinforcement learning (RL) algorithms, where BoT outperforms MLP and vanilla transformer baselines.

- We analyze the computational advantages of BoT, by showing how reformulating the scaled dot product in the computation of the attention operation leads to near-200% runtime and floating point operations (FLOPs) reduction.

## II. BODY TRANSFORMER

Robot learning policies that employ the vanilla transformer architecture as a backbone typically neglect the useful information provided by the embodiment structure. In contrast, here we leverage this structure to provide a stronger inductive bias to transformers, while retaining the representation power of the original architecture. We propose Body Transformer (BoT), which is based on masked attention (see Section B-C), where at each layer in the resulting architecture, a node can only attend to information from itself and its direct neighbors. As a result, information flows according to the graph structure, with the upstream layers reasoning according to local information and the downstream layers pooling more global information from the farther nodes.

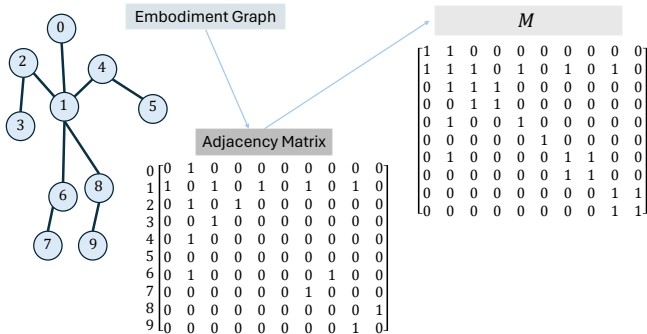

Fig. 2: **Formulation of Embodiment Mask.** The mask $M$ is constructed by adding a diagonal of 1s to the embodiment graph's adjacency matrices. Here, we visualize a simple example of a mask $M$ for an arbitrary agent's embodiment where $n = 10$.

We present below the various components of the BoT architecture (see also Figure 1): (i) a tokenizer that projects the sensory inputs into the corresponding node embedding, (ii) a transformer encoder that processes the input embeddings and generates output features of the same dimension, and (iii) a detokenizer that decodes the features to actions (or values, for RL critic's training).

**Tokenizer.** We map the observation vector to a graph of local observations. In practice, we assign global quantities to the root element of the body, and local quantities to the nodes representing the corresponding limbs, similarly to previous GNN approaches [13, 14, 11, 12]. Then, a linear layer projects the local state vector into an embedding vector. Each node's state is fed into its *node-specific* learnable linear projection, resulting in a sequence of $n$ embeddings, where $n$ represents the number of nodes (or sequence length). This is in contrast to the existing works [14, 11, 12] that use a single *shared* learnable linear projection to deal with varying number of nodes in multi-task RL.

**BoT Encoder.** We use a standard transformer encoder [27] with several layers as a backbone, and present two variants of our architecture:

- *BoT-Hard* masks every layer with a binary mask $M$ that reflects the structure of the graph. Specifically, we construct the mask as $M = I_n + A$, where $I_n$ is the identity matrix of dimension $n$, and $A$ is the adjacency matrix corresponding to the graph (see Figure 2 for an example). Concretely, this allows each node to attend to itself and its direct neighbors, and introduces considerable sparsity in the problem, which is particularly appealing from a computational perspective as highlighted in Section III-D.
- *BoT-Mix* interleaves layers with masked attention (constructed as in BoT-Hard) with layers with unmasked attention. This is similar to the concurrent work in Buterez et al. [3], with the distinctions that, I) we find it more effective in our experimental setting to have a masked attention layer as the first layer, II) our mask $M$ is not equivalent to the adjacency matrix, allowing a node to additionally attend to itself at every layer of the architecture.

**Detokenizer.** The output features from the transformer encoder are fed into linear layers that project them to the actions associated with the node's limb, which are assigned based on the proximity of the corresponding actuator with the limb. Once again, these learnable linear projection layers are separate for each of the nodes. When BoT is employed as a critic architecture in reinforcement learning settings, as in the experiments presented in Section III-B, the detokenizer would output values rather than actions, which are then averaged across body parts.

## III. EXPERIMENTS

We assess the performance of BoT across imitation learning and reinforcement learning settings. We keep the same structure as in Figure 1 and only replace the BoT encoder with the various baseline architectures to single out the effect of the encoder. Particularly, across the various experiments listed in this section, we present the following baselines and variations: (i) an MLP that stacks all embedding vectors as its input, (ii) a vanilla unmasked transformer encoder, (iii) BoT-Hard that only uses masked self-attention layers, (iv) BoT-Mix that alternates between masked and unmasked self-attention layers. All comparisons are made across models with a similar number of trainable parameters.

With the following experiments, we aim to answer the following questions:

- Does masked attention benefit imitation learning in terms of performance and generalization?
- Does BoT exhibit a positive scaling trend compared to a vanilla transformer architecture?
- Is BoT compatible with the RL framework, and what are sensible design choices to maximize performance?
- Can BoT policies be applied to a real-world robotics task?
- What are the computational advantages of masked attention?

### A. Imitation Learning Experiments

We evaluate the imitation learning performance of the BoT architecture in a body-tracking task defined through

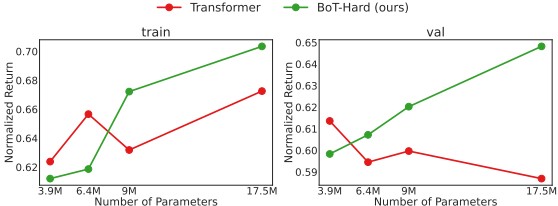

| | normalized episode return | | normalized episode length | |
|---|---|---|---|---|
| | train | val | train | val |
| MLP | 0.821 / 0.586 | 0.739 / 0.540 | 0.957 / 0.765 | 0.915 / 0.726 |
| Transformer | **0.909** / 0.673 | 0.806 / 0.587 | 0.995 / 0.855 | 0.952 / 0.776 |
| BoT-Hard (ours) | 0.908 / **0.703** | **0.890 / 0.648** | **1.000 / 0.876** | **1.000 / 0.841** |
| Multi-Clip [29] | - / 0.654 | - / - | - / 0.855 | - / - |

(a) **Training and Validation Performance Across Architectures.** Statistics of the various architecture-criterion combinations are shown with two values, the leftside being the maximum mean value recorded throughout all evaluation epochs across three seeds, the rightside being the mean of all evaluation scores recorded in last 15 evaluation epochs across all three seeds.

(b) **Training and Validation Performance Across Number of Trainable Parameters.** Each datapoint representing a parameter count-architecture combination is the mean of its throughout the last 15 evaluation epochs across three seeds. We use the 17.5M models for the other imitation learning experiments in this paper.

Fig. 3: **BoT Performance on Imitation Learning.**

the MoCapAct dataset [29], which comprises action-labeled humanoid state trajectories with over 5M transitions, spanning a total of 835 tracking clips. For each architecture, we train a deterministic behavioral cloning (BC) policy. We evaluate mean returns normalized by the length of a clip, in addition to the normalized length of an episode, which terminates when the tracking error goes beyond a threshold. We run the evaluations both on the training and the (unseen) validation clips.

We report results in the table shown in Figure 3a, where BoT consistently outperforms the MLP and transformer baselines. Remarkably, the gap with these architectures further increases on the unseen validation clips, demonstrating the generalization capabilities provided by the embodiment-aware inductive bias. We also report the performance on the training set obtained by a tailored multi-clip policy presented in Wagener et al. [29] with the MoCapAct dataset. While the multi-clip policy is competitive with the vanilla transformer baseline, it is strongly outperformed by our architecture. This is a particularly remarkable result, as the comparison presents conditions more favorable to the baseline, which features a more flexible stochastic policy, was optimized in a recurrent fashion tailored to the tracking task, and was trained on a larger set of rollouts.

As shown in Figure 3b, we also find that BoT-Hard exhibits strong scaling capabilities, as its performance keeps improving with the number of trainable parameters compared to the transformer baseline, both on the training and validation clips. This further indicates a tendency for BoT-Hard to not overfit to the training data, which is induced by the embodiment bias. Additional experiments and comparisons are reported in Section F.

### B. Reinforcement Learning Experiments

We evaluate the RL performance of BoT and baselines using PPO [22] on 4 robotic control tasks in Isaac Gym [15]: Humanoid-Mod, Humanoid-Board, Humanoid-Hill, and A1-Walk.

All humanoid environments are built on top of the classical Humanoid environment in Isaac Gym, where we modify the observation space to increase the number of distributed sensory information (see details in Section C) and include contact forces at all limbs. Humanoid-Mod features the classical run-

ning task on flat ground, while in Humanoid-Hill we replaced the flat ground with an irregular hilly terrain. Humanoid-Board is a newly designed task, where the task is for the humanoid to keep balancing on a board placed on top of a cylinder. Finally, we adapt the A1-Walk environment, which is part of the `legged_gym` repository [20], where the task is for a Unitree A1 quadruped robot to follow a fixed velocity command.

Figure 4 presents the average episode return of evaluation rollouts during training for MLP, Transformer, and BoT (Hard and Mix). The solid curve corresponds to the mean, and the shaded area to the standard error over three seeds. The result shows that BoT-Mix consistently outperforms both the MLP and vanilla transformer baselines in terms of sample efficiency and asymptotic performance, highlighting the efficacy of integrating body-induced biases into the policy network architecture. Meanwhile, BoT-Hard performs better than the vanilla transformer on simpler tasks (A1-Walk and Humanoid-Mod), but shows relatively inferior results in hard-exploration tasks (Humanoid-Board and Humanoid-Hill). Given that the masked attention bottlenecks information propagation from distant body parts, BoT-Hard's strong constraints on information communication may hinder efficient RL exploration: In Humanoid-Board and Humanoid-Hill, it may be useful for information about sudden changes in ground conditions to be transmitted from the toes to the fingertips in the upstream layers. For such tasks, BoT-Mix strikes a good balance between funneling information through the embodiment graph and enabling global pooling at intermediate layers to ensure efficient exploration. In contrast, in A1-Walk or Humanoid-Mod, the environment's state changes more regularly, thus the strong body-induced bias can effectively reduce the search space, enabling faster learning with BoT-Hard.

### C. Real World Experiments

The Isaac Gym simulated locomotion environments are widely popular for sim-to-real transfer of RL policies without requiring adaptation in the real-world [20]. To verify that our architecture is suitable for real-world applications, e.g., running in real time, we deploy one of the BoT policies trained above to a real-world Unitree A1 robot, adapting the codebases in Zhuang et al. [37] and Wu et al. [31]. This is showcased in the video at https://bodytransformer.site, demonstrating the

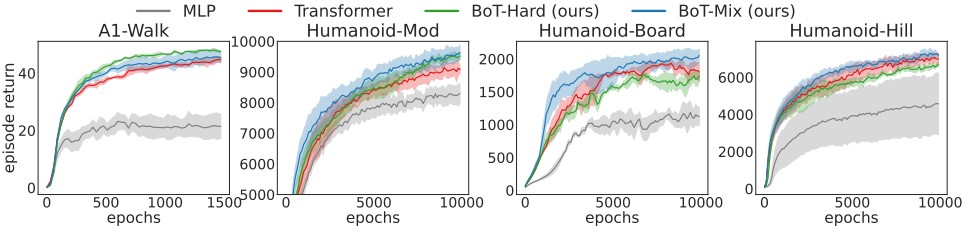

(a) Performance on various tasks across evaluated architectures.

Fig. 4: **Reinforcement Learning Performance on Robotic Control Tasks.**

feasibility of our architecture for real-world deployment. We note that for simplicity we did not make use of teacher-student training or memory mechanisms [16] as common in the locomotion literature, which are both known to further improve the transfer by resulting in more natural gaits.

### D. Computational Analysis

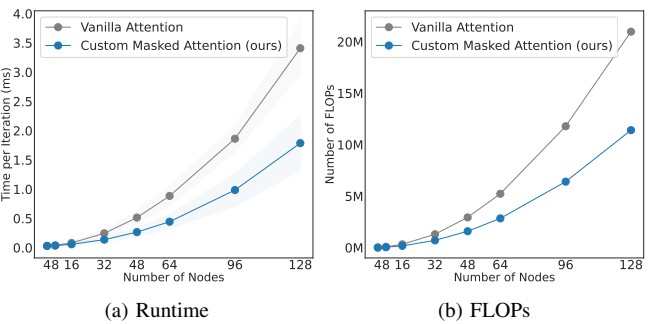

| (a) Runtime | (b) FLOPs |
|---|---|

Fig. 5: **Computational analysis of the custom masked attention implementation.** Across 10,000 randomly sampled masks, we found that our custom implementation provides a 200% speed-up in runtime at sequence lengths up to 128 nodes and scales better in number of FLOPs.

Connections between body parts of a physical agent are often sparse, and so are the pre-computable masks $M$ for embodiment graphs in BoT. Masked attention mechanisms can benefit from this sparsity, as their computational cost can be reduced by ignoring unnecessary computations of those matrix elements that will eventually be masked out. Large-purpose deep learning libraries such as `PyTorch` feature largely optimized matrix multiplication and attention routines (e.g., FlashAttention [6]), but do not leverage possible sparsity in the masked attention mechanism, ascribed by Buterez et al. [3] to missing use cases so far. For a fair computational comparison, we re-implement the scaled dot product in Equation (1) using CPU-based `NumPy` and evaluate on a single sample and attention head, being their batched and multi-head versions further parallelizable on GPUs.

Our custom implementation comprises two major changes in the computation of the masked attention mechanism from its vanilla implementation in `PyTorch`. First of all, we only perform the dot product computation for elements that will not be masked, resulting in fewer matrix multiplication-induced FLOPs from the computation of $\frac{1}{\sqrt{d_k}}QK^T$. Secondly, we only use unmasked values to compute the softmax in Equation (1), also resulting in reduced FLOPs.

We measure the average runtime of each implementation of the attention mechanism across 10,000 set of randomly

generated $Q$, $K$, $V$, and $M$. For each randomization, the generated masks $M$ have a diagonal of 1s and sparsity equal to that used in the MoCapAct experiments ($\beta = 0.908$) [1].

In Figure 5a, we present scaling results of our implementation against vanilla attention across sequence lengths (number of nodes) $\{4, 8, 16, 32, 48, 64, 96, 128\}$. We observe potential speed-ups of up to 206% for 128 nodes (e.g., in the order of humanoids with dexterous hands [26]). Overall, this shows that the body-induced bias in our BoT architecture not only enhances the overall performance of physical agents but also benefits from the naturally sparse masks that the architecture imposes. With adequate parallelization, this approach may significantly reduce the training time of learning algorithms as shown above.

A depiction of the amount of FLOPs required across different sequence lengths is also presented in Figure 5b. Further details and derivation about these experiments can be found in Section I.

### IV. CONCLUSION

In this work, we presented a novel graph-based policy architecture, Body Transformer, which exploits the robot body structure as an inductive bias. Our experiments show how masked attention, which is at the core of BoT, benefits both imitation and reinforcement learning algorithms. Additionally, the architecture exhibits favorable scaling and computational properties, making it appealing for applications on high-dimensional systems.

Here, we used transformers to process sequences of distributed sensory information from the same timestep. However, transformers have been shown to excel at processing information across time too. We leave the extension of BoT to the temporal dimension as future work, as it promises to further improve real world deployment of robot policies, such as the one demonstrated on the Unitree A1 robot.

A limitation of our approach is the fact that its computational advantages are currently not fully exploited by modern deep learning libraries, and we hope that this work may stimulate future developments in this direction. In addition, our architecture requires a minimum amount of transformer layers to make sure that the architecture does not lose representation power in modeling long-range relations, which generally increases the amount of required trainable parameters.

---

[1] A mask with sparsity $\beta$ has $\beta n^2$ zero elements. When $\beta = 1 - \frac{1}{n}$, the mask reduces to the identity $I_n$. In practice, the maximum degree of a vertex (i.e. node) in robots will be approximately constant, making the computational complexity of masked attention grow linearly with the number of nodes.

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

# APPENDIX A
## RELATED WORK

**Transformers in robotics.** Originally developed for NLP applications [27], transformers have been successfully applied across domains, for example in computer vision [8] and audio processing [7]. Several works have shown applications of transformers as a means to represent robot policies [35, 4, 36], demonstrating its core advantages in this setting, i.e., variable context length, handling long sequences [19] and multiple modalities [18, 25]. However, these approaches use transformers as originally developed for unstructured or grid-like inputs, such as language or images, respectively. In this work, we leverage the robot embodiment by appropriately adapting the transformer attention mechanism.

**Graph Neural Networks (GNNs).** GNNs [32] are a class of learning architectures that can process inputs in the form of a graph [21]. While early versions of these architectures featured explicit message-passing schemes along the graph [2, 10], more recent architectures mostly feature attention-based approaches. In fact, the vanilla transformer, with its variable context length, inherently supports fully connected graphs. However, state-of-the-art performance on graph interpretation benchmarks is only achieved via modifications of the original transformer architecture, for example by means of learned graph encodings and attention biases [34]. A contemporaneous work, Buterez et al. [3], following a similar idea as in the work from Veličković et al. [28], utilizes masked attention layers, where each node only attends to its neighbors, and interleaves such layers with unmasked attention layers. In this work, we exploit masked attention in a policy learning setting, by additionally proposing an architecture that only comprises layers where each can attend to itself and its direct neighbors, resulting in naturally growing context over layers, i.e., the outputs of the first layers are computed using more local information compared to those of the last layers.

**Exploiting body structure for policy learning.** Graph neural networks have been explored by several works as a way to obtain multi-task RL policies that are effective across different robot morphologies. Earlier works focused on message passing algorithms [30, 13], and were later outperformed by vanilla transformers [14, 11] and transformer-based GNNs that make use of learned encoding and attention biases [12]. All these approaches were only demonstrated in simulated benchmarking scenarios and not applied to a real-world robotics setting. Compared to previous work, we additionally show that introducing bottlenecks in the attention mask fully exploits the embodiment structure and benefits policy learning also for tasks achieved by a single agent, leading to better performance and more favorable scaling.

# APPENDIX B
## BACKGROUND

### A. Attention Mechanisms in Transformers

Transformer, a foundational architecture in modern machine learning applications as well as in our work, is powered by the self-attention mechanism [27]. Self-attention weighs the values corresponding to each element of the sequence with a score that is computed from pairs of keys and queries extracted from the same sequence. Thus, it is able to identify relevant pairs of sequence elements in the model output.

Concretely, the self-attention output vector is computed through the following matrix operation:

$$\text{Attention}(Q, K, V) = \text{softmax}\left(\frac{QK^T}{\sqrt{d_k}}\right)V,$$

where $Q$, $K$, $V$ (respectively query, key, and value matrices) are learnable linear projections of the sequence elements' embedding vectors, and $d_k$ is the dimension of the embedding space. As embedding pairs with higher correspondence will have a higher score (from the computation of $QK^T$), the corresponding value vector of an embedding's associated key will receive a higher weight in the attention mechanism output.

## B. Transformer-based GNNs

In this work, we model the agent embodiment as a graph whose nodes are sensors and actuators, and their connecting edges reflect the body morphology. While message-passing GNNs are suitable inductive biases for this formulation, they tend to suffer from oversmoothing and oversquashing of representations, preventing effective long-range interactions and discouraging network depth [1].

More recently, self-attention was proposed as an alternative to message-passing [34, 14]. While the standard self-attention mechanism amounts to modeling a fully connected graph, a popular transformer-based GNN, Graphormer [34], injects node-edges information through graph-based positional encodings [17, 33, 34, 12] and by biasing the scaled dot-product, i.e.,

$$\text{Attention}(Q, K, V) = \text{softmax}\left(\frac{QK^T}{\sqrt{d_k}} + B\right)V, \quad (1)$$

where $B$ is a learnable matrix that depends on graph features, e.g., shortest path or adjacency matrix, and can effectively encode the graph structure in a flexible manner.

## C. Masked Attention

The attention mechanism can be altered [27] with a binary mask $M \in \{0,1\}^{n \times n}$ (where $n$ is the sequence length), which is equivalent to replacing the elements of $B$ in (1):

$$B_{i,j} = \begin{cases} 0 & M_{i,j} = 1 \\ -\infty & M_{i,j} = 0 \end{cases},$$

where $i$ and $j$ denote row and column indices. This operation effectively results in zeroing out the contribution of the pairs indicated with zeros in the mask $M$ to the computation of the attention.

### APPENDIX C
### DETAILS ON RL ENVIRONMENTS

We adapt the IsaacGym humanoid environment for the three humanoid-related tasks, by modifying the observation space to include the vertical position of the torso, root coordinates and angular velocity, joint positions and velocities, and per-limb contact forces. We leave the reward for the Humanoid-Mod and Humanoid-Hill unchanged, while we adapt the reward for Humanoid-Bob by forcing the forward target velocity to zero, and appropriately adjusting the target and termination heights to take the balancing board into account. For the A1-Walk task, we adapt the codebase in Zhuang et al. [37] and train the policies using proprioception only for the actor, and additional simulation parameters for the critic. We define the task to mantain a target velocity of 0.5 $m/s$ on an irregular terrain.

### APPENDIX D
### POSITIONAL ENCODINGS

For the reinforcement learning experiments presented in Section III-B, we found that the use of positional encodings improves the performance of BoT architectures. Specifically, we compute positional encodings through an embedding layer

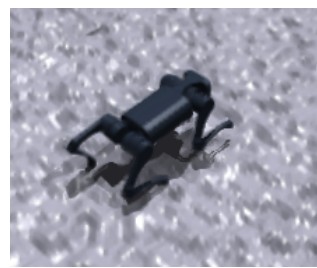 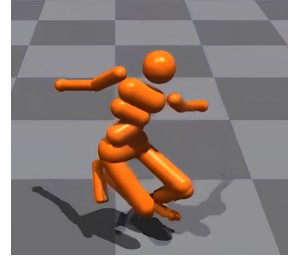

(a) Unitree A1-Walk        (b) Humanoid-Mod

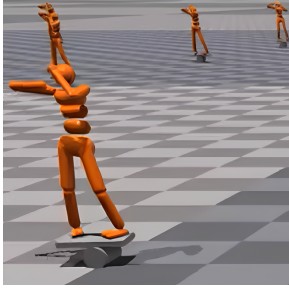 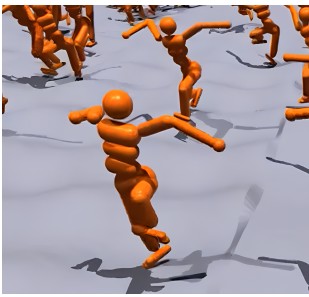

(c) Humanoid-Board        (d) Humanoid-Hill

Fig. 6: Snapshots of successful rollouts of BoT policies.

that maps indices – up to $n$ – to encoding vectors, which are then added to the tokenizers' outputs. While this is beneficial for the reinforcement learning setting, we did not report a considerable improvement in the imitation learning setting, which we present without the use of positional encodings. In fact, these are not strictly necessary, as in the BoT architecture tokenizers do not share weights across body parts, and may in principle replace the role of positional encodings.

### APPENDIX E
### REAL-WORLD DEPLOYMENT

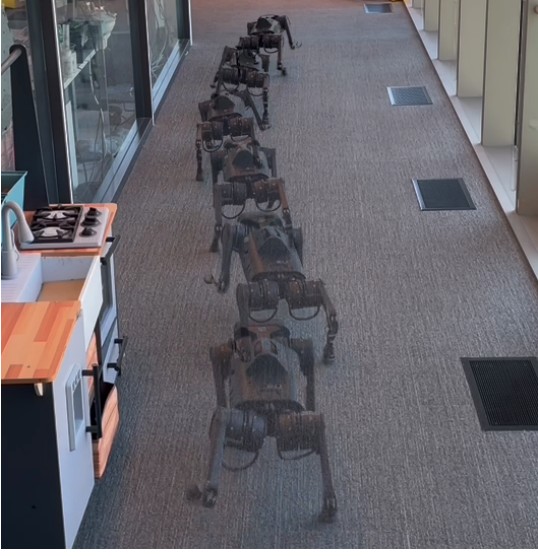

Fig. 7: **Real-World Deployment.** Frame overlay demonstrating the deployment of the BoT walking policy to a Unitree A1 quadruped robot.

We deployed the RL policy trained for A1-Walk task to

a real-world Unitree A1 Robot. Computation was runned offboard on CPU and commands were sent via WiFi or Ethernet connection. The video at https://bodytransformer.site demonstrates the real-world deployment. A frame overlay representing the robot motion are also shown in Figure 7.

## APPENDIX F
### ADDITIONAL IMITATION LEARNING ABLATIONS

In this section we provide several ablations in addition to those presented in Section III-A. Specifically, we compare (i) BoT-Hard, (ii) BoT-Mix, (iii) BoT-Soft, which – similarly to [12] – learns the matrix $B$ in (1) as a function of the graph's shortest path matrix, and (iv) BoT-Hard/Random with a randomly sampled mask, i.e. having ones on its diagonal and the same sparsity as the mask $M$ used for the correct implementation of BoT-Hard.

The table in Figure 8 shows the result of this comparison, with BoT-Hard outperforming all baselines on most of of the metrics. The bottlenecks introduced by the masked attention result in better performance compared both to a mixed approach (BoT-Mix) and an approach that also accounts for structure but does not prevent long-range communication (BoT-Soft). As expected, simply sampling a random mask without properly accounting for the embodiment structure deteriorates performance.

## APPENDIX G
### ADDITIONAL REINFORCEMENT LEARNING ABLATIONS

#### A. Effect of Body-Induced Masking in BoT

BoT relies on masked attention with its mask determined by the embodiment structure. We conduct an additional experiment in the RL setting to further demonstrate the effect of the body-induced masking in this setting. We compare with BoT-Hard/Random and BoT-Mix/Random, where the attention mask $M$ is given by a randomly sampled symmetric binary matrix with the same degree of sparsity ($\beta \approx 0.82$ for the IsaacGym humanoid). The results are presented in Figure 9. Overall, BoT with *random* masking (dotted lines) underperforms BoT with *body-induced* masking (solid lines) in both a simpler task (Humanoid-Mod) and a hard-exploration task (Humanoid-Board), which highlights that the use of body-induced masking is crucial for the performance of BoT.

#### B. Effect of Per-Limb Tokenizer vs. Shared Tokenizer

The existing works using Transformer-based policies [14, 11, 12] for multi-task RL adopt *shared* linear projections for tokenizers and detokenizers to deal with the varying number of limbs, i.e., per-limb observation features are projected into embedding vectors by the single shared tokenizer network, and the per-limb hidden vectors are transformed to per-limb actions via the single shared detokenizer network. In contrast, our BoT is designed for tasks with a single morphology, thus we adopt *per-node* linear projections for tokenizer and detokenizer. We conduct an additional experiment to investigate the effect of this design choice, and the results are demonstrated in Figure 10.

In Figure 10, the solid lines denote the results of using per-node tokenizers/detokenizers, and the dotted lines present the results of using a shared tokenizer/detokenizer (which can be understood as representatives of the existing methods [14, 11, 12]). Overall, Transformer/BoT with per-node (de)tokenizers significantly outperform their shared (de)tokenizer counterparts in both a simpler task (Humanoid-Mod) and a hard-exploration task (Humanoid-Board). This shows that the use of tokenizers shared across different limbs for Transformer-based policies hinders efficient learning.

## APPENDIX H
### TRAINING DETAILS

The training parameters of the experiments detailed in Section III-A and Section III-B are as summarized in Tables 11a and 11b. We run ten evaluation rollouts for both training and validation sets after each training epoch.

## APPENDIX I
### FLOP DERIVATION FOR CUSTOM MASKED ATTENTION IMPLEMENTATION

Below, we comparatively analyze an asymptotic bound for the amount of floating-point operations required in one scaled dot product (see Equation (1)) call between the vanilla and the masked approach. From hereon, let $n$ denote the sequence length and $d_{\mathrm{k}}$ the input and output dimension of our attention mechanism.

**Computing** $\frac{QK^T}{\sqrt{d_k}}$. Considering $Q \in \mathbb{R}^{n \times d_{\mathrm{k}}}$ (and similarly for $K$), the computation of $QK^T$ will generally require $d_{\mathrm{k}}$ multiplications and $d_{\mathrm{k}}-1$ additions for all of $n^2$ pairs. Division by $\sqrt{d_k}$ results in $n^2$ divisions and one constant factor $c_1$ of FLOPs for computing $\sqrt{d_k}$. The total amount of flops is $2n^2 d_{\mathrm{k}} + c_1$.

**Masked computation of** $\frac{QK^T}{\sqrt{d_k}}$. Exploiting sparsity, we ignore all inner product computations for zero entries in $M$, computing only $\beta n^2$ pairs of multiplications. This results in a reduction to $2\beta n^2 d_{\mathrm{k}} + c_1$ FLOPs.

**Computing** $\mathrm{Softmax}(S)$. A softmax for one vector of dimension $n$ requires $n$ exponentiations, $n - 1$ additions, and $n$ divisions, performed for $n$ rows. Let exponentiations require $c_2$ FLOPs per element, then a total of $(2 + c_2)n^2 - n$ FLOPs is performed.

**Masked computation of** $\mathrm{Softmax}(S)$. As a result of sparsity, there is instead a total of $\beta n^2$ exponentiations $\beta n^2$ divisions, and $\beta n^2 - n$ additions to compute, reducing our demand to $(2 + c_2)\beta n^2 - n$ FLOPs.

**Computing the multiplication** $\mathrm{Softmax}(S)V$. A total of $nd_{\mathrm{k}}$ pairs are multiplied, where each pair requires $2n - 1$ operations to complete. The total amount of FLOPs is $2n^2 d_{\mathrm{k}} - nd_{\mathrm{k}}$. Following a similar reasoning with previous writing, a total of $2n^2 d_{\mathrm{k}} - nd_{\mathrm{k}}$ FLOPs are performed.

Assuming that our physical agent provides a graph-induced mask $M \in \{0, 1\}^{n \times n}$ of sparsity $\beta \in [\frac{1}{n}, 1]$ (such that there are $\beta n^2 > n$ nonzero entries), then the amount of FLOPs required by a vanilla masked self-attention implementation is $4n^2 d_{\mathrm{k}} + (2 + c_2)n^2 - nd_{\mathrm{k}} - n + c_1$, while that of a custom masked implementation is $(2\beta + 2)n^2 d_{\mathrm{k}} + (2 + c_2)\beta n^2 - nd_{\mathrm{k}} - n + c_1$. Therefore, the performance gap between the vanilla and masked implementations is determined by the sparsity

| | Return Normed | | Length Normed | |
|---|---|---|---|---|
| | train | test | train | test |
| BoT-Hard (ours) | 0.908 / **0.703** | **0.890 / 0.648** | **1.000 / 0.876** | **1.000 / 0.841** |
| BoT-Mix (ours) | **0.943** / 0.679 | 0.844 / 0.604 | 0.982 / 0.853 | 0.964 / 0.785 |
| BoT-Soft | 0.900 / 0.678 | 0.843 / 0.598 | 0.993 / 0.859 | 0.962 / 0.789 |
| BoT-Hard/Random | 0.850 / 0.661 | 0.835 / 0.600 | 0.995 / 0.845 | 0.962 / 0.782 |

Fig. 8: **Additional Imitation Learning Ablations.** Statistics of the various architecture-criterion combinations are shown with two values, the leftside being the maximum mean value recorded throughout all evaluation epochs across three seeds, the rightside being the mean of all evaluation scores recorded in last 15 evaluation epochs across all three seeds.

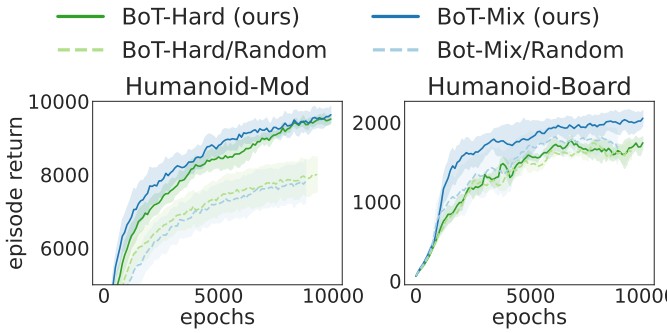

Fig. 9: **Additional RL Experimental Results on the Effect of Body-induced Masking.**

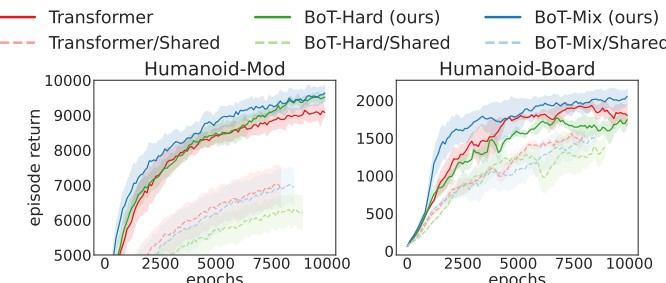

Fig. 10: **Additional RL Experimental Results on the Effect of Per-Node (De)Tokenizers.**

| Parameter | Values | |
|---|---|---|
| | MLP | Transformers |
| Batch Size | 256 | 256 |
| # Epochs | 100 | 100 |
| # Encoder Layers | 3 | 16 |
| Embedding Input Size | 320 | 320 |
| Feedforward Size | 2500 | 1024 |
| # Attention Heads | N/A | 5 |
| Learning Rate | 1e-4 | 1e-4 |
| # Parameters | 16,696,656 | 17,544,120 |

(a) **Training Parameters Used for Imitation Learning Experiments.**

| Parameter | Values | |
|---|---|---|
| | MLP | Transformers |
| Num Envs | 2048 | 2048 |
| Batch Size | 8192 | 8192 |
| # Encoder Layers | 3 | 10 |
| # Attention Heads | N/A | 2 |
| Embedding Input Size | N/A | 64 |
| Feedforward Size | 150 | 128 |
| # Parameters | 699,467 | 688,762 |

(b) **Training Parameters Used for Reinforcement Learning Experiments.**

Fig. 11: **Training Parameters Used for Experiments in Section III.**

coefficient $\beta$, that is, the number of FLOPs that a vanilla approach requires will be $c(n)$ times the number of FLOPs a custom masked approach requires:

$$\lim_{n\to\infty} \frac{\text{\# FLOPs in vanilla approach}}{\text{\# FLOPs in masked implementation}}$$
$$= \lim_{n\to\infty} \frac{4n^2 d_{\text{k}} + (2 + c_2)n^2 - n d_{\text{k}} - n + c_1}{(2\beta + 2)n^2 d_{\text{k}} + (2 + c_2)\beta n^2 - n d_{\text{k}} - n + c_1}$$
$$= \frac{4 d_{\text{model}} + 2 + c_2}{(2\beta + 2) d_{\text{model}} + 2\beta + \beta c_2} \geq 1$$

Therefore, even though these implementations share the same asymptotic bound $\mathcal{O}(n^2 d_{\text{k}})$, the custom masked implementation's amount of FLOPs scales better than the vanilla implementation.

Note that it is possible to further optimize our implementation by sparsifying the multiplication $\text{Softmax}(S)V$; this is left as a direction of future work, and requires the use of sparse array libraries, which was not in the scope of this analysis.