# OpenReview forum: "Body Transformer: Leveraging Robot Embodiment for Policy Learning"
_roboticsfoundation.org/RSS/2024/Workshop/EARL — EARL 2024 Poster_

### Official Review · Reviewer_MY8j · 2024-06-24

**Rating:** 6
**Confidence:** 4

**Review:**

The authors propose to use masked attention in the transformer architecture in order to encode a robots embodiment into the architecture. The idea intuitively makes sense and shows some promising performance based on the author's evaluations. It is very close in spirit to GNN based approaches.
The paper is well written and easy to understand, therefore, I vote for acceptance.

I think in future work, the authors should use more seeds to strengthen their claim. In my opinion 3 seeds are not enough. Furthermore, a more diverse set of environments could be used. In the current version the authors use different versions of humanoids and one experiment on the A1. More classical Mujco/DMC environments would strengthen their analysis.

Two minor points:
- Fig 3. a+b) should contain confidence intervals
- Fig 4. Does not need a caption and sub caption, since there is only one plot.

---

### Decision · Program_Chairs · 2024-06-24

Accept (Poster)